# Measuring real-time disease transmissibility with temperature-dependent generation intervals

**Esther Li Wen Choo**[1], **Kris V. Parag**[2], **Jo Yi Chow**[1], **Jue Tao Lim**[1]*

**1** Lee Kong Chian School of Medicine, Nanyang Technological University, Singapore, Singapore, **2** MRC Centre for Global Infectious Disease Analysis, Imperial College London, London, United Kingdom

* juetao.lim@ntu.edu.sg

## Abstract

Accurate real-time estimation of the effective reproduction number ($R_t$) is critical for infectious disease surveillance and response. In vector-borne diseases like dengue, temperature strongly influences disease transmission by affecting generation times. However, most existing $R_t$ estimation methods assume a fixed generation interval, leading to biased estimates and unreliable assessments of transmission risk in settings with fluctuating temperatures. In this study, we proposed and evaluated a novel framework to estimate a temperature-dependent reproduction number (td-$R_t$) that dynamically updates the generation interval distribution based on observed temperature data. We obtained real-time estimates of td-$R_t$ through an adapted Bayesian recursive filtering process. Using real and simulated data for a temperature-sensitive disease (dengue), we evaluated the performance of td-$R_t$ against the typically used temperature-independent reproduction number (ti-$R_t$) and angular reproduction number ($\Omega_t$), which does not require specification of the generation interval. Simulated data was generated under varying patterns of underlying $R_t$ and temperature datasets. Performance was evaluated by classification accuracy, defined by the proportion of instances where estimated $R_t$ correctly identified whether the true $R_t$ was above or below 1. We found that td-$R_t$ generally outperformed ti-$R_t$ and $\Omega_t$ in classifying periods of epidemic growth. td-$R_t$ achieved the highest classification accuracy in 54 of 72 simulation scenarios, with accuracy ranging from 37.1%–95.9%. td-$R_t$ accuracy was highest in scenarios with greater temperature variability, surpassing other methods by up to 20%. With Singapore dengue case data, td-$R_t$ and $\Omega_t$ signals showed 75% similarity, highlighting $\Omega_t$'s potential as a complementary measure that is less sensitive to model assumptions. These findings highlight the importance of accounting for temperature in real-time transmissibility estimates, as temperature-driven variations in generation time can introduce model misspecification and bias. Incorporating temperature is especially crucial for climate-sensitive diseases like dengue. Future work could extend this framework to other pathogens and additional transmission-relevant covariates.

**Data availability statement:** All code and temperature data used for analysis can be found in https://github.com/EstherChoo/temp-dep-rt. The dengue case data underlying the results presented in the study could not be made publicly available as it is property of the Ministry of Health of Singapore and included geo-referenced dengue case data containing identifying or sensitive patient information. All inquiries for data access should be sent to luis. ponce@nhghealth.com.sg.

**Funding:** This research is funded and supported by the Lee Kong Chian School of Medicine – Ministry of Education Start-Up Grant to JTL. This research / project is also supported by the Ministry of Education, Singapore, under its Academic Research Fund Tier 1 (RT4/22) and Academic Research Fund Tier 1 Seed Funding Grant (RS04/22) to JTL. The funders had no role in study design, data collection and analysis, decision to publish, or preparation of the manuscript.

**Competing interests:** The authors have declared that no competing interests exist.

## Author summary

Many public health decisions rely on the effective reproduction number ($R_t$), the average number of new infections arising from a typical case at a given time. Standard estimators of $R_t$ typically assume a fixed generation interval, which is the time between a primary and a secondary infection. For dengue, that interval shifts with temperature - warmer days shorten virus development inside mosquitoes and accelerate transmission. We developed a real-time estimator that updates the generation interval distribution daily using observed temperatures, then computes $R_t$ from reported cases.

We evaluated this temperature-dependent estimator against two comparators: conventional estimator that holds the interval fixed and an interval-free statistic (the angular reproduction number). In simulations spanning temperature regimes, our method most often correctly classified transmission as growing or controlled (best in 54 of 72 scenarios), especially when temperatures varied widely. Applied to dengue in Singapore, the interval-free statistic showed potential as a complementary measure that is less sensitive to model assumptions.

By embedding temperature into estimation, we reduce bias from misspecified generation intervals and provide a practical, real-time tool for climate-sensitive infections. This framework can support timely and proportionate control decisions and is readily adaptable to other pathogens influenced by environmental conditions.

## Introduction

Understanding and quantifying pathogen transmissibility is fundamental to epidemiological modelling and public health decision-making. A key metric of transmissibility is the effective reproduction number, $R_t$, which represents the average number of secondary cases generated over the period an individual remains infectious [1]. Epidemic growth can be assessed with $R_t$, where an $R_t$ above 1 signals a growing outbreak, which requires timely intervention to prevent further spread [1].

Several statistical algorithms have been developed to estimate $R_t$ such as the Wallinga and Teunis (WT) method, EpiEstim, EpiNow2 and EpiFilter [2–5]. However, these approaches usually assume a known and fixed generation time, which is the duration between successive infections and cannot be observed. This makes the estimation of $R_t$ susceptible to misspecification if the generation interval distribution is incorrect or time-varying [6]. Consequently, public health officials may be misled by the misspecified $R_t$ values, resulting in unnecessary interventions or failure to act during an actual epidemic. Nonpharmaceutical interventions were found to shorten the serial interval of SARS-CoV-2, the duration between symptom onset of successive cases, further providing evidence that assuming a fixed generation interval could

bias the estimates of $R_t$ [7]. Several methods have been formulated to address misspecified generation interval distributions in the estimation of transmissibility. For example, the angular reproduction number measures time-varying changes in the disease transmissibility estimate without requiring any generation time measurements [6]. Another study presented a renewal equation framework for vector-borne diseases that estimates time-varying transmissibility [8].

In temperature-sensitive infectious diseases like dengue, biological processes underlying transmission are known to vary with temperature. Specifically, the extrinsic incubation period of dengue is biologically known to be temperature-dependent in the dengue vector species, *Aedes aegypti* and *Aedes albopictus* [9]. The extrinsic incubation period of dengue is longer at cooler temperatures, due to slower rates of viral replication within the vector [9,10]. However, the methods commonly employed to estimate $R_t$ do not currently incorporate the influence of temperature on transmissibility, which can significantly influence transmission dynamics. Accounting for this temperature dependence in the transmission model could enhance the accuracy and biological plausibility of $R_t$ estimates. This was demonstrated by Codeço et al. who developed a method which considered temperature-dependent generation interval to estimate the $R_t$ of dengue [11]. The study found that incorporating temperature in the specification of the generation interval provided a more precise and accurate estimate of the reproduction number than a temperature-independent $R_t$ [11]. However, this was based on the WT method, which probabilistically links each case to potential infectors using the generation interval distribution. From these infector–infectee pairs, it reconstructs the likely transmission network and epidemic trajectory. This requires information on cases that occur after the time of inference, hence cannot be used to obtain real-time estimates and is more suited for retrospective analysis. To address these gaps, we extended the framework of the EpiFilter method, which can be used for real-time estimation of $R_t$ [4]. EpiFilter is a recursive Bayesian smoother which uses forward filtering and backward smoothing to produce stable and improved $R_t$ estimates [4]. The filtering step uses only past incidence and can therefore provide reliable real-time estimates, while the smoothing step incorporates future incidence with respect to past estimates to refine those estimates as the later data had emerged. As future information is required, the smoothing step is applied to refine estimates across all timepoints except the most recent one. As a result, EpiFilter is more statistically robust at low case incidence than previously developed methods [4].

There is currently no methodology that estimates the reproduction number, $R_t$, in real-time while accounting for temperature effects on disease transmission. Hence, we aim to develop a novel modelling framework that dynamically integrates temperature data to improve real-time $R_t$ estimation. We also aim to assess the accuracy of temperature-dependent $R_t$ estimates in comparison with other existing estimation methods. We first developed a novel framework which incorporated a temperature-dependent serial interval distribution into the EpiFilter algorithm. We then utilised this framework to estimate the temperature-dependent $R_t$, along with the temperature-independent $R_t$ which uses the standard assumption of a temperature-independent generation interval distribution, and angular reproduction number, on dengue case counts in Singapore and compared the estimates of disease transmissibility across methods. Next, we estimated the three types of transmissibility estimates using multiple sets of simulated dengue case data, each generated from different pre-determined $R_t$ trajectories. This allowed us to assess and compare the accuracy and robustness of the estimates. We hypothesise that the temperature-dependent $R_t$ will consistently agree with the true $R_t$ around the epidemic threshold of 1, more than either the temperature-independent $R_t$ or the angular reproduction number.

## Methods

### Data

Daily mean temperature in Singapore was obtained from Meteorological Service Singapore (MSS) [12]. MSS collected daily measurements of each climate variable at 63 automated weather stations, which we averaged to obtain the national average for Singapore. Daily dengue case data arranged by the date of illness onset from 2012 to 2024 was obtained from the Ministry of Health, Singapore [13]. All available dengue case data, including the period overlapping with the

COVID-19 pandemic, were retained to ensure that transmissibility estimates reflected real-world conditions, including potential effects of non-pharmaceutical interventions. Dengue is a notifiable disease in Singapore under the Infectious Diseases Act which requires mandatory reporting of all laboratory-confirmed cases to the Ministry of Health [14]. Confirmatory diagnostic testing for dengue is widely available in Singapore. Rapid diagnostic tests for dengue are commonly used in primary care, while additional tests such as PCR and ELISA are available in hospitals [15,16]. Cases confirmed by rapid diagnostic tests are included in the dataset and are not always subsequently confirmed by PCR or ELISA.

**Temperature-dependent generation interval distribution of dengue**

The generation interval distribution, which characterizes the distribution of the time between infections, is a key component in estimating $R_t$. As the generation interval cannot be observed, we approximated it with the serial interval which is the time between symptom onset between cases. The serial interval was derived by aggregating the duration of each stage in the dengue transmission cycle. Each stage of the dengue transmission cycle was modelled as in Table 1. As *Aedes aegypti* is the primary and most efficient vector of dengue, the transmission stages were parameterized based on this species. Temperature-dependence was incorporated into the extrinsic incubation period as the rate of viral replication within the mosquito vector is strongly influenced by ambient temperature, whereas the other stages in the transmission cycle (i.e., intrinsic incubation period, human-to-mosquito transmission and mosquito-to-human transmission) are less directly affected by temperature [9,17]. We estimated both a temperature-independent and temperature-dependent $R_t$ to evaluate whether incorporating temperature improves $R_t$ estimation. The temperature-dependent extrinsic incubation period was used in the temperature-dependent serial interval distribution while the temperature-independent extrinsic incubation period was used in the temperature-independent serial interval distribution. The modelling approaches of both temperature-independent and temperature-dependent $R_t$ is described in the sections below.

We first modelled the intrinsic incubation period, $I_H$, the time from infection to symptom onset, with a gamma distribution, which was found in previous literature to be a good approximation of the intrinsic incubation period [10]. The time for human-to-mosquito transmission, $T_{HM}$, was assumed to follow an exponential distribution which reflected the rapid decline in transmission probability after the first 3 days of illness [18]. The temperature-dependent extrinsic incubation period, $I_M^*$, was modelled using a gamma distribution, with temperature incorporated as a covariate of the rate parameter. This formulation, as demonstrated by Chan and Johansson, provided a good fit to data compiled from multiple empirical studies [10]. The temperature-independent extrinsic incubation period, $I_M$, was assumed to be an exponential distribution with the extrinsic incubation rate of 0.23, which was also demonstrated to be a good fit by Chan and Johansson in a temperature-independent approximation [10]. Following Codeço et al, the mosquito to human transmission time, $T_{MH}$, was modelled as an exponential distribution, with the most transmission activity taking place in the first 3–4 days of infectiousness [11]. This follows the assumption that there is a limited window of a few days to transmit the virus to humans as infected mosquitoes would spend the majority of their lifespan in the incubation phase.

**Table 1. Parameterisation of serial interval distribution.**

| Dengue Transmission Cycle | Probability Distribution |
| --- | --- |
| Intrinsic incubation period | $I_H \sim Gamma(shape = 16, rate = 2.7)$ |
| Human to mosquito transmission | $T_{HM} \sim exp(1)$ |
| Extrinsic incubation period (temperature-independent) | $I_M \sim exp(0.23)$ |
| Extrinsic incubation period (temperature-dependent) | $I_M^* \sim Gamma(shape = 4.3, rate = 7.9 - 0.21T)$, where T (°C) is the contemporaneous temperature |
| Mosquito to human transmission | $T_{MH} \sim exp(1)$ |

The temperature-independent serial interval distribution, $w = \{w_u\}_{u=1}^{U}$, and temperature-dependent serial interval distribution, $w(T) = \{w_u(T)\}_{u=1}^{U}$, can be expressed as the convolution of the four distributions as in Eq. (1) and (2). The convolution of gamma distributions was computed through a recursive formula in [19]. The use of $w_u$ to estimate reproduction number is described in the sections below.

$$w = I_H + T_{HM} + I_M + T_{MH} \tag{1}$$

$$w(T) = I_H + T_{HM} + I_M{}^* + T_{MH} \tag{2}$$

### Estimating the effective reproduction number for temperature-sensitive infectious diseases

The renewal model framework encapsulates the relationship between disease incidence and the effective reproduction number under the assumption of a homogenous and well-mixed population. Specifically, the observed incidence, $I_t$, is assumed to be Poisson distributed, conditioned on the effective reproduction number, $R_t$, and total infectiousness, $\Lambda_t$, at time $t$:

$$I_t \sim Poisson(\Lambda_t R_t), \quad \Lambda_t = \sum_{u=1}^{U} I_{t-u} w_u, \tag{3}$$

$\Lambda_t$ represents total infectiousness, which is the expected number of infections that can be generated at time $t$ by previously infected individuals. The serial interval distribution, $w = \{w_u\}_{u=1}^{U}$, as specified in the section above, is used to weight prior incidence $I_{t-u}$ when estimating total infectiousness, $\Lambda_t$. Each $w_u$ represents the probability that a primary infection occurring $u$ days prior resulted in a secondary infection at time $t$, for each time point $u = 1, 2, ..., U$. The distribution satisfies $\sum_{u=1}^{U=\infty} w_u = 1$ but was truncated to $U = 35$ for computational efficiency, which captures over 99% of the probability mass across all temperature ranges used in the analysis. The same definition was applied for the temperature-dependent serial interval distribution, $w(T) = \{w_u(T)\}_{u=1}^{U}$.

We estimated $R_t$ based on the past incidence curve using the EpiFilter algorithm, which was adapted from Bayesian recursive filtering [4]. EpiFilter was adopted as it could generate real-time estimates and use all data available to the modeller at the contemporaneous timepoint. To compare the utility of incorporating temperature dependence, we estimated two versions of $R_t$ with EpiFilter: a novel temperature-dependent reproduction number (td-$R_t$) and a conventional temperature-independent reproduction number (ti-$R_t$). In EpiFilter, first, a state model is used to address the autocorrelation between reproduction numbers, characterised by the following equation:

$$R_t = R_{t-1} + (\eta \sqrt{R_{t-1}})\epsilon_{t-1} \tag{4}$$

$R_t$ is taken to be a hidden Markov state to be inferred, which dynamically depends on the previous state $R_{t-1}$ and a noise term $\epsilon_{t-1} \sim N(0, 1)$. The noise term is scaled by a fraction, $\eta < 1$, of $\sqrt{R_{t-1}}$. $R_t$ is assumed to take a discrete value within a closed space, $\mathcal{R} := \{R_{min}, R_{min} + \delta, ..., R_{max}\}$, for some $R_{min}, R_{max}$ and grid space, $\delta$.

The estimation of $R_t$ begins with recursive filtering, which involves a prediction and filtering step. In the prediction step, a sequential prior predictive distribution, $p_t{}^*$, is constructed based on past incidence, $I_1^{t-1}$, and the previous state, $R_{t-1}$, as in Eq. (5a). Next, the filtering step updates the prior predictive distribution into a posterior filtering distribution based on the latest observation, $I_t$, in Eq. (5b).

$$p_t{}^* = P\left(R_t \middle| I_1^{t-1}\right) = \int P\left(R_t \middle| R_{t-1}, I_1^{t-1}\right) p_{t-1} \, dR_{t-1} \tag{5a}$$

$$p_t \propto P\left(I_t \middle| R_t, I_1^{t-1}\right) p_t{}^* \tag{5b}$$

where $P\left(R_t|R_{t-1}, I_1^{t-1}\right) \sim N(R_{t-1}, \eta^2 R_{t-1})$ following the state equation in Eq (4) and $P\left(I_t|R_t, I_1^{t-1}\right) \sim Poisson(\Lambda_t R_t)$ following the observation equation in Eq (3). In our novel temperature-dependent framework, we incorporated temperature-dependence in $w_u$ as described in the section above, which was then used to estimate $P\left(I_t|R_t, I_1^{t-1}\right)$ in Eq. (5b).

Recursive filtering is followed by the recursive backward smoothing step to update the posterior filtering distribution as new data accumulates.

$$q_t = p_t \int P\left(R_{t+1}|R_t, I_1^t\right) q_{t+1} p_{t+1}^{-1} \, dR_{t+1}$$

(6)

where $p_t = P\left(R_t|I_1^t\right)$ is the filtering distribution and $p_{t+1} = P\left(R_{t+1}|I_1^t\right)$ is the predictive distribution obtained in Eq (5). $P\left(R_{t+1}|R_t, I_1^t\right) \sim N\left(R_t, \eta^2 R_t\right)$ is from the state equation in Eq (4). Eq (6) is solved by noting that $q_t = p_t$ and iterating backwards in time to obtain the first smoothing distribution $q_1$. The integrals become sums over the grid $\mathcal{R}$ and distributions are the element vectors in the grid. The point estimates of $R_t$ were obtained from the posterior mean of $q_t$, while the 95% Bayesian credible intervals of the estimated $R_t$ were computed directly from the 2.5th and 97.5th quantiles of $q_t$. EpiFilter is a deterministic algorithm and exact for a given prior and grid over the space of $R_t$, unlike sample-based approaches. Consequently, multiple runs at the same settings on the same data always produce the same posterior estimate.

1-step ahead incidence was obtained through the posterior predictive distribution by solving the integral over the grid $\mathcal{R}$:

$$P\left(I_{t+1}|I_1^t\right) = \int P\left(I_{t+1}|I_1^t, R_t\right) q_t \, dR_t$$

(7)

Following [20], it is assumed that $P\left(I_{s+1}|I_1^s, R_s\right) \sim Poisson(\Lambda_{t+1} R_t)$. Similarly, the 95% Bayesian credible intervals of future incidence were computed directly from the 2.5th and 97.5th quantiles of the posterior predictive distribution. The 1-step ahead predictions were assessed by their mean absolute percentage error.

## Estimating transmissibility without generation interval

The generation interval is necessary for most transmissibility estimation algorithms. However, it involves assumptions about the timing of transmission events that may not hold in all settings and may be prone to misspecification. By incorporating temperature in the estimation of $R_t$, we incorporate a key component of transmission dynamics into the modelling framework and ideally improve the accuracy of the estimated $R_t$ signals. In addition, we sought to compare the temperature-dependent reproduction number to a metric that addresses this limitation by not requiring the specification of the generation interval altogether while still responding to changes in the generation time distribution.

The angular reproduction number, $\Omega_t$, defines transmissibility as a ratio of new infections to M, the root mean square of past infections over a user-defined window. This removes the need for knowledge of generation times. $\Omega_t$ responds to both changes in $R_t$ and the generation time distribution. The renewal model in Eq (3) is re-defined as a function of $M_t$ and $\Omega_t$ as follows:

$$I_t \sim Poisson(\Omega_t M_t), \quad M_t = \left(\frac{1}{\delta} \sum_{u=t-\delta}^{t-1} I_u^2\right)^{1/2}$$

where we set $\delta = 24$ as it is heuristically recommended to be twice or thrice the mean generation interval time [6]. Within this recommended range, the estimation of $\Omega_t$ is robust to the choice of $\delta$ [6]. The full derivation of $\Omega_t$ can be found in Text A in S1 File. $\Omega_t$ can be interpreted similar to $R_t$, where a value above 1 signals a growing epidemic and a value below 1 signals a waning epidemictl.

$\Omega_t$ was estimated using the EpiFilter algorithm as described in the previous section by replacing the usual input of total infectiousness, $\Lambda_t$, with the root mean square of past infections, $M_t$. The algorithm outputs the complete posterior

distribution $P\left(\Omega_t|I_1^N,\ \delta\right)$, where $N$ is the length of data used. The mean estimate $\hat{\Omega}_t$ and 95% credible intervals were obtained from the posterior distribution like before. The 1-step ahead prediction was obtained from the posterior predictive distribution $P\left(I^t|I_1^{t-1},\ \delta\right)$ given by the EpiFilter algorithm.

## Simulation study

To measure the absolute accuracy of the transmissibility estimates, we estimated td-$R_t$, ti-$R_t$ and $\Omega_t$ on simulated dengue data generated from pre-determined sets of $R_t$. We let the true underlying $R_t$ be a sine curve to represent periodic patterns in transmission dynamics. We then calculated the true $\Lambda_t$, as in Eq (3), assuming the true generation interval of dengue to be temperature dependent. Under the renewal framework, the number of new dengue cases at time $t$ was modelled as a Poisson random variable with rate parameter given by the product of the $R_t$ and $\Lambda_t$, which encapsulates past incidence weighted by the serial interval distribution:

$$I_t \sim Poisson\left(\Lambda_t R_t\right),\quad \Lambda_t = \sum_{u=1}^{t-1} I_{t-u} w_u(T)$$

where $w_u(T)$ is the temperature-dependent generation interval distribution and T is the contemporaneous temperature in °C. To investigate the effects of different underlying $R_t$ values on the accuracy of the estimated $R_t$, we varied the amplitude, period and smoothness of the underlying $R_t$ values used to generate the case data. Smoothness was modified by adding random noise to the $R_t$ values. Furthermore, we used four different sets of temperature data in the different simulation scenarios, which were incorporated into the generation interval distribution of dengue when generating case data and estimating td-$R_t$. The four datasets included daily temperature data in Singapore in 2012, daily temperature data in upper Northern Thailand (Chiang Mai, Lamphun, Lampang, Uttaradit, Phrae, Nan, Phayao, Chiang Rai, Mae Hong Son) in 2021, and two synthetic temperature datasets with more variance than Singapore's temperature, which were normally distributed with a mean of 28 and standard deviation of 1.2 and 2 respectively. The Singapore temperature dataset had the smallest temperature range from 24.6°C – 30.0°C, followed by the two synthetic temperature datasets which ranged from 25.1°C – 31.2°C and 24.5°C – 32.7°C. The Northern Thailand temperature dataset had the widest temperature range of 18.0°C – 29.0°C. This allowed us to examine the performance of td-$R_t$ under different temperature scenarios, specifically in settings with varying degrees of temperature fluctuations.

A total of 72 simulated case datasets were generated by combining three amplitude levels of $R_t$, three periodicities of $R_t$, smooth versus non-smooth $R_t$ dynamics, and four temperature datasets. The synthetic cases were simulated to reflect different plausible ranges of outbreak dynamics to ensure our results were relevant to real-world dengue epidemics. In large-scale outbreaks such as the 2024 dengue outbreak in Brazil, case counts can rise to several thousand per day [21]. Meanwhile in smaller countries, dengue cases can increase from about 50 cases a day to 200 daily cases over a span of 3 months, as observed in the Singapore 2020 dengue outbreak [22]. The simulated cases generally display one major outbreak within the 200-day period. In these scenarios, case counts fluctuate over time but follow an overall increasing trend, which is consistent with the pattern typically observed in real dengue outbreaks. In other scenarios, the simulations show smaller fluctuations without a distinct peak, which may represent minor or well-controlled outbreaks. Some scenarios feature a single peak, resembling outbreaks commonly observed in Singapore, while others display multiple peaks, similar to patterns seen in larger or more seasonally variable countries such as Thailand or Brazil. Our simulated case datasets were therefore constructed to capture this range of epidemic trajectories, allowing us to test model performance across diverse but realistic outbreak scenarios.

We estimated td-$R_t$, ti-$R_t$ and $\Omega_t$ on the 72 sets of simulated case data and compared their percentage accuracy with the respective true, underlying $R_t$. $\Omega_t$ is a different transmissibility statistic from td-$R_t$ and ti-$R_t$, however the three types of

estimates could be similarly interpreted based on whether they were above or below 1, indicating a growing or waning epidemic respectively. Percentage accuracy was calculated based on the proportion of time in the simulation for which the true $R_t$ and transmissibility estimate were both above or below 1, considering only timepoints where the estimates were statistically significant based on the 95% credible interval. To avoid bias from edge effects, we excluded the first window, $\delta$ = 24 days, of estimates. To have a more complete evaluation of the metrics' performance, the classification accuracy of the transmissibility estimate was also measured by the AUC-ROC metric, which quantifies how well the estimates were able to discriminate between positive and negative classes—specifically, whether the transmissibility estimate was below or above the critical threshold of 1, indicating controlled or growing transmission, respectively.

To assess the robustness of our transmissibility estimates, we evaluated their classification accuracy when the transmission rate, and in turn the generation interval, was misspecified. We generated four distinct groups of simulations with varying degrees of mismatch between the true and assumed transmission rate. Each group comprised of 54 simulated epidemic curves generated from $R_t$ values with three amplitude levels, three periodicities, two types of $R_t$ dynamics (smooth versus non-smooth), and three temperature datasets (Singapore temperature and two higher-variance synthetic temperature datasets). Two groups of simulations were based on a generation interval that underestimated transmission rate while two groups were based on a generation interval that overestimated transmission rate. This mismatch between the true and assumed transmission dynamics allowed us to evaluate how sensitive the estimation framework is to misspecification of the generation interval, which may be driven by exogenous changes not captured by the model, such as period of intensified or relaxed vector control efforts which change the contact rate of humans and mosquitoes. The mismatch could also result from changes in transmission driven by other environmental factors such as precipitation or humidity. Case data was generated with the same parameters in Table 1, with modification to $T_{HM} \sim \exp(\rho)$, $T_{MH} \sim \exp(\rho)$, $\rho := \{0.25,\ 0.33,\ 3,\ 4\}$ for each of the four groups. Overestimating transmission rate by 4 and 3 times shortens the generation interval by about 2 and 1 day respectively. Conversely, underestimating transmission rate by 4 and 3 times lengthens the generation interval by about 5 and 3 days respectively. A lower rate $\rho$ represents reduction in transmission rate due to a decreased mosquito population from vector control interventions, whereas a higher value reflects an increased transmission rate that could occur if vector control measures were removed. Similar to the simulations above, the transmissibility estimates were compared by their percentage accuracy and the AUC-ROC metric.

## Application to dengue data in Singapore

To test their applicability in practical settings, we estimated the temperature-dependent reproduction number (td-$R_t$), temperature-independent reproduction number (ti-$R_t$) and angular reproduction number ($\Omega$) on daily dengue data from 2012 to 2024 in Singapore, where dengue is endemic. To estimate the temperature-dependent reproduction number (td-Rt), we computed the serial interval distribution as a function of contemporaneous daily temperature in Singapore. We then compared the estimates to assess their similarities under real-world data. The estimates were compared pairwise by percentage agreement. Percentage agreement was calculated based on the proportion of transmissibility estimates which were both above or below 1, or were both not statistically significant. For each transmissibility estimate, we obtained 1-step ahead dengue predictions from the posterior mean of the predictive distribution and evaluated forecast accuracy by the mean absolute percentage error (MAPE) between the predicted and observed dengue cases.

We also investigated the effect of underreporting on the transmissibility estimates, where cases with recent onset may not have been reported yet. We assumed reporting delays to be lognormally distributed, with a mean reporting delay of 3 days and standard deviation of 3 days. This places most of the probability mass between 1–5 days, with a long tail to represent rare cases of long reporting delays up to 14 days. Each reported case in Singapore from 2022-2024 was assigned a reporting delay by adding a randomly drawn delay to the onset date to generate a simulated reporting date. Case counts with reporting dates beyond the latest timepoint were removed, which simulates right truncation and under-estimation of recent cases which have yet to be reported. We iteratively obtained the real-time

estimates of td-$R_t$, ti-$R_t$ and $\Omega_t$ by re-running EpiFilter at each day using only data available up to that point, accounting for reporting delays.

## Results

In our framework, we incorporated temperature into the parameterisation of the extrinsic incubation period and consequently the serial interval distribution, which was used as an input in the EpiFilter algorithm to estimate td-$R_t$. The temperature-dependent serial interval distribution was much narrower and more concentrated than the temperature-independent serial interval distribution (Fig 1). The serial interval became shorter as temperature increases, ranging from a mode of 11.5 days at 33 °C to 8 days at 37 °C (Fig 1).

We estimated $\Omega$, td-$R_t$ and ti-$R_t$ using simulated dengue case counts generated under varying conditions, including different amplitudes, periods, and smoothness levels of the true underlying $R_t$ (Fig A in S1 File). Furthermore, we simulated dengue case counts assuming a temperature-dependent generation interval, using temperature datasets with varying degrees of variability—real daily temperature data from Singapore, upper Northern Thailand and two synthetic datasets with increasing variance (Fig B in S1 File). We calculated the percentage accuracy of the transmissibility estimates according to the proportion of timepoints where they agree with the true underlying $R_t$ with respect to the threshold of 1, i.e., both estimated and true $R_t$ were below and above 1. Overall, td-$R_t$ had the best accuracy, outperforming $\Omega_t$ and ti-$R_t$ in 54 out of 72 scenarios and tying with ti-$R_t$ in 1 of the scenarios (Table 2). ti-$R_t$ had the next best accuracy, which performed the best in 13 of the simulations (Table 2). The strong performance of td-$R_t$ was corroborated by AUC-ROC values, which measured the classification ability of the $R_t$ estimates. $R_t > 1$ was considered a positive class while $R_t \leq 1$ was considered a negative class. td-$R_t$ had the best AUC-ROC value in 56 out of 72 simulations (Fig C in S1 File). In the other 16

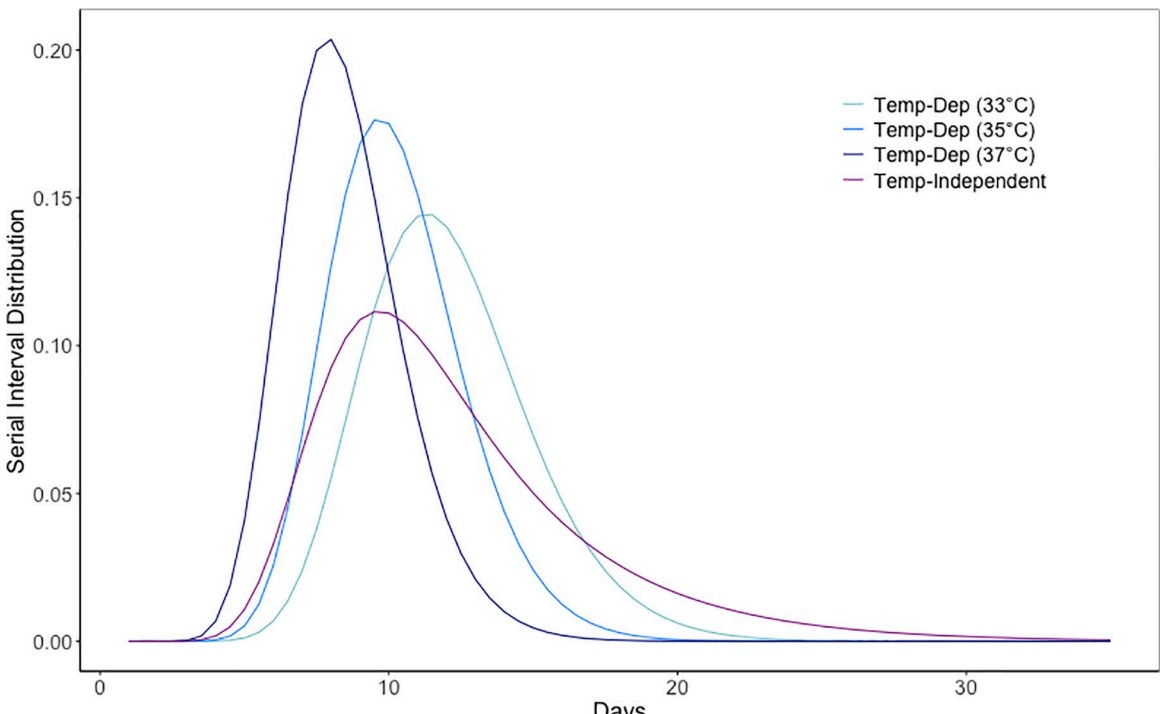

**Fig 1. Temperature-independent and temperature-dependent serial interval distributions at 33 °C, 35 °C and 37 °C respectively.**

**Table 2. Percentage accuracy of estimated angular reproduction number ($\Omega_t$), temperature-independent reproduction number (ti-$R_t$) and temperature-dependent reproduction number (td-$R_t$) under simulations with varying period, amplitude and smoothness of true underlying $R_t$ and varying temperature datasets. Percentage accuracy was calculated based on whether the significant transmissibility estimates and true Rt were both below 1 or above 1. The transmissibility estimate with the highest accuracy in each scenario was underlined.**

| Noisy $R_t$ | | | | | | Smooth $R_t$ | | | | | |
|---|---|---|---|---|---|---|---|---|---|---|---|
| Period | Amplitude | Temperature | $\Omega_t$ (%) | td-$R_t$ (%) | ti-$R_t$ (%) | Period | Amplitude | Temperature | $\Omega_t$ (%) | td-$R_t$ (%) | ti-$R_t$ (%) |
| 0.05 | 1 | Singapore data | 41.2 | 54.1 | 64.7 | 0.05 | 1 | Singapore data | 61.8 | 72.9 | 74.7 |
| 0.1 | 1 | Singapore data | 52.4 | 67.1 | 60 | 0.1 | 1 | Singapore data | 64.1 | 74.7 | 57.1 |
| 0.15 | 1 | Singapore data | 42.4 | 43.5 | 33.5 | 0.15 | 1 | Singapore data | 67.1 | 75.3 | 61.8 |
| 0.05 | 2 | Singapore data | 88.8 | 88.2 | 87.6 | 0.05 | 2 | Singapore data | 81.8 | 82.9 | 84.1 |
| 0.1 | 2 | Singapore data | 77.6 | 88.2 | 88.2 | 0.1 | 2 | Singapore data | 74.7 | 83.5 | 82.4 |
| 0.15 | 2 | Singapore data | 74.7 | 85.3 | 84.1 | 0.15 | 2 | Singapore data | 69.4 | 83.5 | 81.2 |
| 0.05 | 3 | Singapore data | 91.2 | 95.3 | 92.9 | 0.05 | 3 | Singapore data | 95.3 | 96.5 | 94.1 |
| 0.1 | 3 | Singapore data | 87.6 | 93.5 | 90 | 0.1 | 3 | Singapore data | 84.1 | 92.4 | 89.4 |
| 0.15 | 3 | Singapore data | 90.6 | 92.4 | 92.9 | 0.15 | 3 | Singapore data | 88.2 | 91.2 | 90.6 |
| 0.05 | 1 | More Variance | 48.2 | 58.8 | 65.9 | 0.05 | 1 | More Variance | 70.6 | 77.6 | 70 |
| 0.1 | 1 | More Variance | 57.1 | 64.1 | 55.3 | 0.1 | 1 | More Variance | 70 | 72.4 | 52.9 |
| 0.15 | 1 | More Variance | 38.8 | 37.1 | 34.1 | 0.15 | 1 | More Variance | 62.4 | 78.8 | 65.3 |
| 0.05 | 2 | More Variance | 87.6 | 90.6 | 88.2 | 0.05 | 2 | More Variance | 82.4 | 87.1 | 88.2 |
| 0.1 | 2 | More Variance | 78.8 | 85.9 | 82.9 | 0.1 | 2 | More Variance | 74.7 | 82.9 | 80.6 |
| 0.15 | 2 | More Variance | 84.1 | 88.8 | 85.9 | 0.15 | 2 | More Variance | 69.4 | 81.8 | 78.8 |
| 0.05 | 3 | More Variance | 92.4 | 93.5 | 93.5 | 0.05 | 3 | More Variance | 94.7 | 95.9 | 93.5 |
| 0.1 | 3 | More Variance | 88.2 | 91.8 | 90.6 | 0.1 | 3 | More Variance | 85.9 | 91.8 | 89.4 |
| 0.15 | 3 | More Variance | 91.2 | 92.9 | 90.6 | 0.15 | 3 | More Variance | 87.1 | 91.8 | 90 |
| 0.05 | 1 | Most Variance | 36.5 | 61.8 | 68.8 | 0.05 | 1 | Most Variance | 76.5 | 78.8 | 75.9 |
| 0.1 | 1 | Most Variance | 43.5 | 64.7 | 55.9 | 0.1 | 1 | Most Variance | 58.8 | 68.8 | 64.1 |
| 0.15 | 1 | Most Variance | 40 | 54.1 | 43.5 | 0.15 | 1 | Most Variance | 56.5 | 75.9 | 58.2 |
| 0.05 | 2 | Most Variance | 83.5 | 84.7 | 87.6 | 0.05 | 2 | Most Variance | 89.4 | 90 | 88.8 |
| 0.1 | 2 | Most Variance | 81.8 | 85.9 | 84.7 | 0.1 | 2 | Most Variance | 72.9 | 79.4 | 79.4 |
| 0.15 | 2 | Most Variance | 75.9 | 85.3 | 83.5 | 0.15 | 2 | Most Variance | 74.1 | 85.9 | 81.8 |
| 0.05 | 3 | Most Variance | 94.7 | 94.7 | 92.9 | 0.05 | 3 | Most Variance | 94.7 | 94.7 | 92.4 |
| 0.1 | 3 | Most Variance | 87.1 | 91.2 | 86.5 | 0.1 | 3 | Most Variance | 87.1 | 90.6 | 87.6 |
| 0.15 | 3 | Most Variance | 91.2 | 90.6 | 89.4 | 0.15 | 3 | Most Variance | 87.6 | 91.2 | 90 |
| 0.05 | 1 | North Thai Data | 39.4 | 65.3 | 67.1 | 0.05 | 1 | North Thai Data | 56.5 | 74.1 | 73.5 |
| 0.1 | 1 | North Thai Data | 47.1 | 62.4 | 51.2 | 0.1 | 1 | North Thai Data | 57.6 | 73.5 | 62.9 |
| 0.15 | 1 | North Thai Data | 30 | 25.9 | 30 | 0.15 | 1 | North Thai Data | 55.9 | 70 | 63.5 |
| 0.05 | 2 | North Thai Data | 85.9 | 90.6 | 91.2 | 0.05 | 2 | North Thai Data | 71.2 | 80 | 79.4 |
| 0.1 | 2 | North Thai Data | 83.5 | 87.6 | 84.7 | 0.1 | 2 | North Thai Data | 70.6 | 81.8 | 78.8 |
| 0.15 | 2 | North Thai Data | 84.7 | 85.9 | 85.9 | 0.15 | 2 | North Thai Data | 72.4 | 82.9 | 80.6 |
| 0.05 | 3 | North Thai Data | 95.3 | 95.9 | 93.5 | 0.05 | 3 | North Thai Data | 92.9 | 94.7 | 94.1 |
| 0.1 | 3 | North Thai Data | 94.1 | 92.4 | 89.4 | 0.1 | 3 | North Thai Data | 90 | 91.2 | 91.8 |
| 0.15 | 3 | North Thai Data | 90.6 | 89.4 | 92.4 | 0.15 | 3 | North Thai Data | 91.2 | 92.4 | 91.8 |

simulations, the AUC-ROC value was only marginally smaller than the next best estimate (Fig D in S1 File). Furthermore, $\Omega_t$ outperformed ti-$R_t$ in 54 out of 72 simulations (Fig C in S1 File).

We used different temperature datasets to define the generation interval distribution, which was then used to generate simulated dengue case counts. The Singapore temperature dataset exhibited the least variation, ranging from 24.6 °C to

30.0 °C, while the Northern Thailand dataset had the most variation, ranging from 18.0 °C to 29.0 °C (Fig B in S1 File). Based on the daily temperature of Northern Thailand, we estimated the generation time to have a mean of 16.7 days and variance of 2.49 days. Meanwhile, based on temperature in Singapore, the generation time was estimated to have a mean of 14.9 days and variance of 0.66 days. Despite these differences in temperature patterns, td-$R_t$ consistently showed stronger accuracy across all temperature scenarios, ranging from 25.9% to 95.9% accuracy (Table 2). On the other hand, adding random noise to the underlying $R_t$ of the simulated case counts potentially affected the performance of td-$R_t$. Without random noise, td-$R_t$ had the best accuracy in 33 out of 36 simulations with an accuracy of 70.0% to 96.5% (Table 2). With random noise, td-$R_t$ had the best accuracy in 21 out of 36 simulations with an accuracy of 25.9% to 95.9% (Table 2). While $\Omega_t$ generally performed worse than td-$R_t$ and ti-$R_t$, it was comparatively less poor when the underlying $R_t$ was noisy rather than smooth.

Generally, $\Omega_t$, td-$R_t$ and ti-$R_t$ estimated by dengue case counts generated a true $R_t$ with a lower amplitude had a lower absolute accuracy (Table 2). $R_t$ estimated on data with a true $R_t$ with a amplitude of 1 and period of 0.05 had an accuracy of 54.1% to 78.8%, whereas $R_t$ estimated on data with a true $R_t$ with a amplitude of 3 and period of 0.05 had an accuracy of 91.2% to 96.5% (Table 2). In Fig 2, we narrowed our analysis down to scenarios where the amplitude of $R_r$ was 2, which reflected realistic estimates of $R_t$, and where underlying $R_t$ was smooth to facilitate clearer visualisation. This allowed us to see how varying the temperature data set and the period of the true underlying $R_t$ affected the accuracy of the transmissibility estimates. When temperature and underlying $R_t$ fluctuates more, the difference in accuracy is largest between estimated td-$R_t$ (75.9%) and the other 2 $R_t$ estimates (58.2% & 56.5%) (Fig 2I). When the temperature range was wide, as with the Northern Thailand temperature dataset, td-$R_t$ had an accuracy of 73.5% while td-$R_t$ and $\Omega_t$ had an accuracy of 62.9% and 57.6% respectively (Fig 2K). In contrast, where temperature is more stable as in the Singapore dataset (Fig 2A) or where the underlying $R_t$ fluctuates less (Fig 2A, 2D, 2G and 2J), the difference in accuracy between td-$R_t$ and ti-$R_t$ or $\Omega_t$ is less pronounced, though td-$R_t$ still demonstrates higher accuracy. The full version of Fig 2 with all simulation scenarios can be found in Fig D in S1 File.

To test the robustness of the estimates under a misspecified generation interval, we simulated case data under scenarios with and without vector control interventions, which were deliberately excluded from the transmissibility estimation process. In the presence of vector control interventions, when transmission rate between human and mosquito was assumed to be overestimated by 3 and 4 times, ti-$R_t$ had the best percentage accuracy in 32 and 34 out of 54 simulations respectively (Tables A and B in S1 File). Based on the AUC-ROC values, ti-$R_t$ and $\Omega_t$ were generally able to best classify periods of epidemic growth when transmission rate was overestimated (Table E in S1 File). Meanwhile, when the transmission rate between human and mosquito was underestimated by 3 and 4 times, td-$R_t$ had the best percentage accuracy in 44 and 45 out of 54 simulations respectively (Tables C and D in S1 File). However, based on the AUC-ROC values, ti-$R_t$ had the best classification performance in the most number of simulations (Table E in S1 File).

We estimated the temperature-dependent reproduction number (td-$R_t$), temperature-independent reproduction number (ti-$R_t$) and angular reproduction number ($\Omega$) on dengue data from Singapore across 2012–2024. Daily dengue cases in Singapore ranged from 0 to 224, averaging at about 25 cases a day (Fig 3A) The mean generation times were derived from the temperature-dependent and temperature-independent generation interval distributions, respectively (Fig E in S1 File). The temperature-independent generation time was fixed at 10 days, while the temperature-dependent generation time varied over time, fluctuating between 13 and 18 days (Fig E in S1 File). We considered the three Rt estimates to be aligned when they simultaneously exceeded or fell below the epidemic threshold of 1. Estimates which were both not statistically significant were also regarded to be concordant. Overall, $\Omega_t$ performed more similarly to td-$R_t$ than to ti-$R_t$. The overall percentage agreement between td-$R_t$ and $\Omega_t$ was 75.0% while the percentage agreement between ti-$R_t$ and $\Omega_t$ was 48.0% (Table 3 and Fig 3B and 3C). The overall percentage agreement between td-$R_t$ and ti-$R_t$ was 64.2% (Table 3).

We obtained 1-step ahead predictions from the mean of the posterior predictive distribution for each reproduction number and evaluated them by the mean absolute percentage error (MAPE) between the predicted and observed dengue

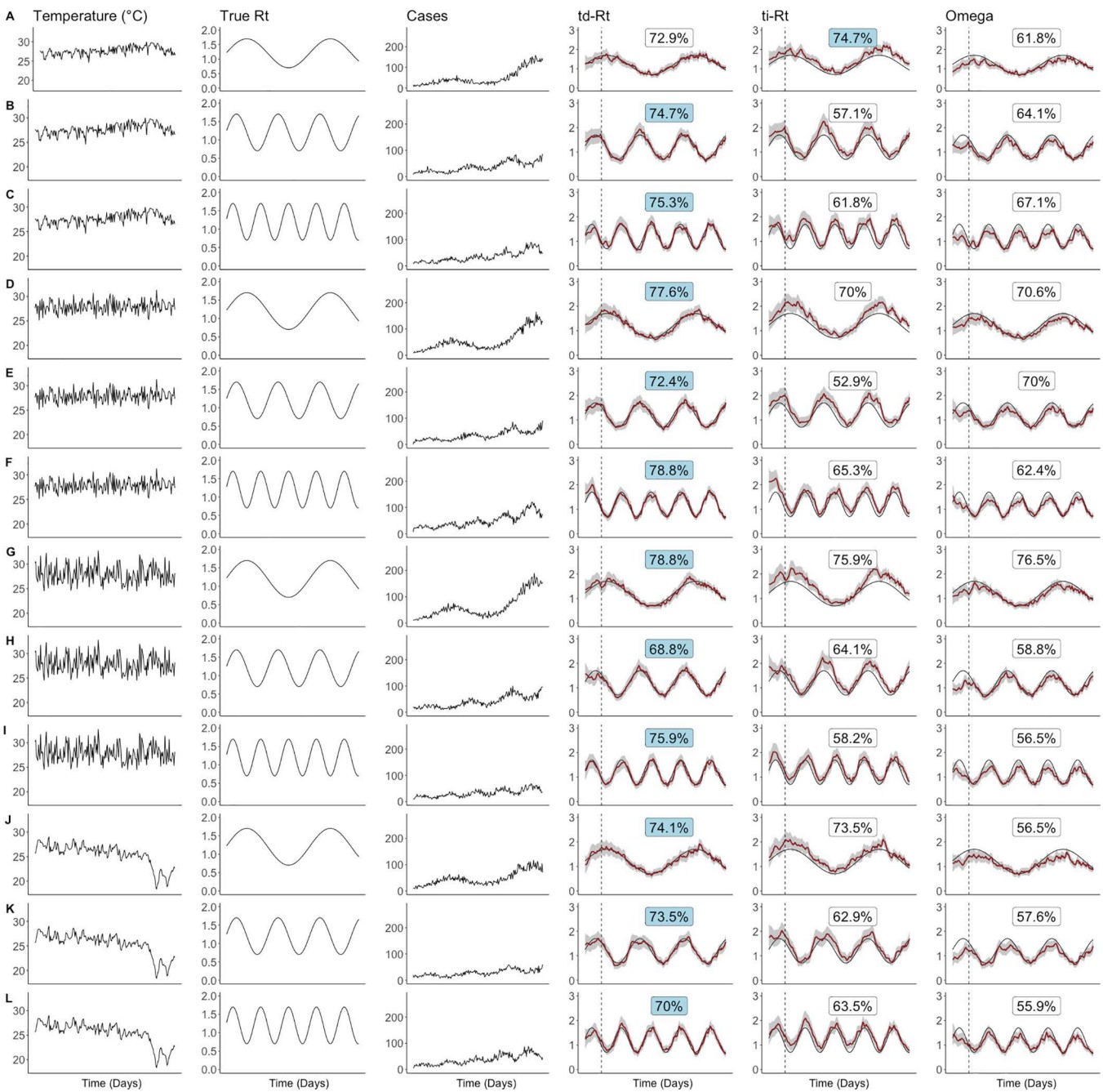

**Fig 2. Estimated angular reproduction number (Omega), estimated temperature-independent reproduction number (ti-Rt) and estimated temperature-dependent reproduction number (td-Rt) under simulation with varying temperature datasets, true underlying reproduction number (Rt) and cases.** Each row represents a set of simulation, which is run over 200 days. The values in the td-Rt, ti-Rt and Omega columns represent the percentage agreement by the threshold of 1 with the true Rt. **(A)**, **(B)**, **(C)**: Singapore daily temperature; **(D)**, **(E)**, **(F)**: Synthetic daily temperature data with more variation; **(G)**, **(H)**, **(I)**: Synthetic daily temperature data with most variation; **(J)**, **(K)**, **(L)**: Upper Northern Thailand daily temperature. The best-performing estimate is indicated by a blue label. In the td-Rt, ti-Rt, Omega columns, the black lines represent the true values, the red lines represent estimated values and shaded areas represent the 95% credible interval. Classification accuracy was calculated excluding the initial period of 24 days, indicated by the dashed line, to avoid edge effects.

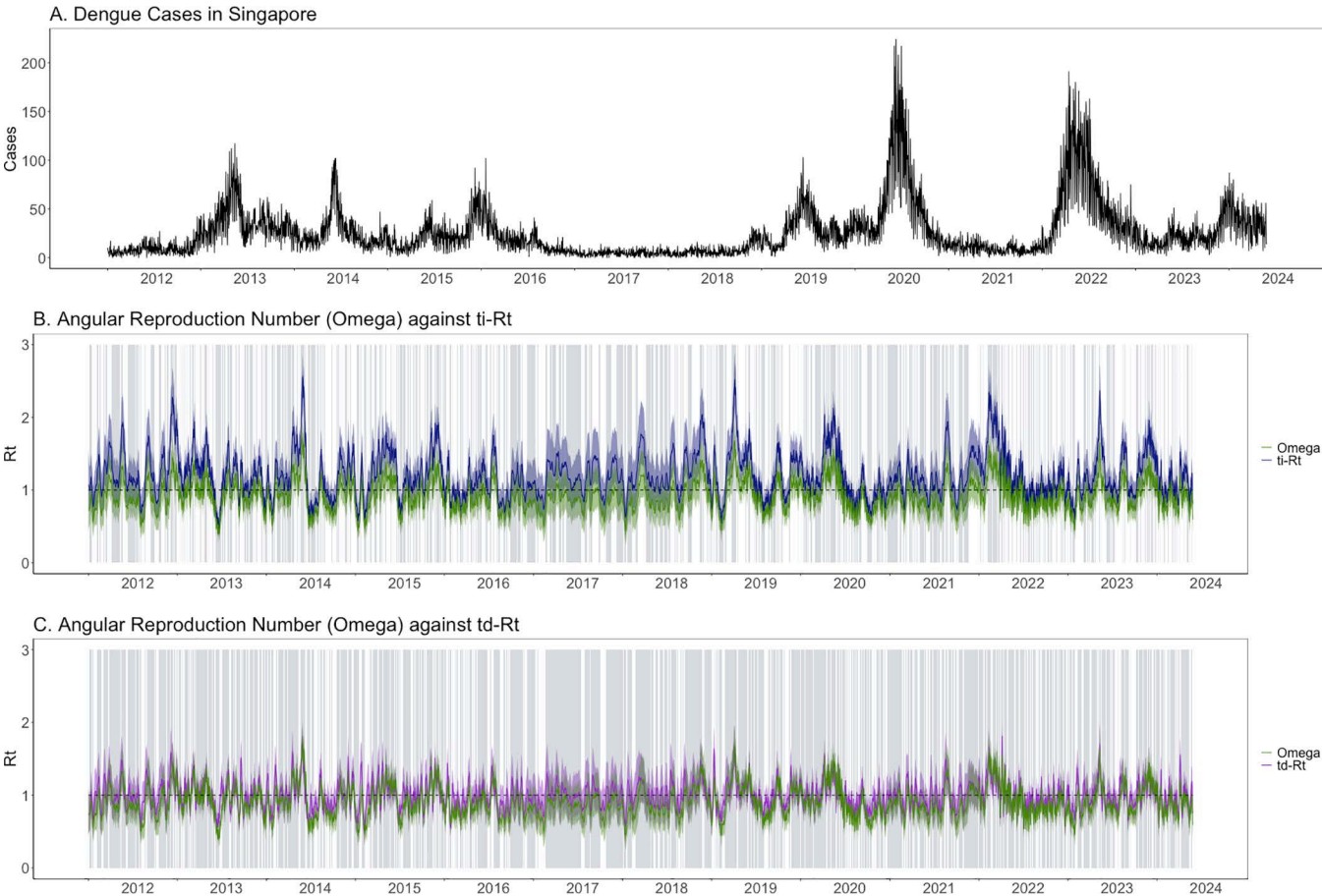

**Fig 3. (A) Dengue cases in Singapore, (B) angular reproduction number against temperature-dependent reproduction number (td-R$_t$) and (C) angular reproduction number against temperature-independent reproduction number (ti-R$_t$) estimated from Singapore dengue cases from 2012 to 2024.** Grey shaded areas represent timepoints where the respective pair of R$_t$ have concordant signals.

**Table 3. Percentage agreement of angular reproduction number ($\Omega$), temperature-dependent reproduction number (td-R$_t$) and temperature-independent reproduction number (ti-R$_t$).**

|  | Both ≤ 1 or Both > 1 (%) | Both not significant (%) | Total (%) |
|---|---|---|---|
| td-R$_t$/ $\Omega_t$ | 19.0 | 56.0 | 75.0 |
| td-R$_t$/ ti-R$_t$ | 16.1 | 48.1 | 64.2 |
| ti-R$_t$/ $\Omega_t$ | 15.5 | 32.5 | 48.0 |

cases. The MAPE of predictions generated by $\Omega$, td-R$_t$ and ti-R$_t$ were 86%, 87.6% and 87.1% respectively, indicating negligible differences in predictive accuracy among the three methods (Fig F in S1 File). This is because prediction performance is generally insensitive to the choice of generation time if the R$_t$ estimate and the corresponding total infectiousness used for estimation, $\Lambda_t$, are used together to generate predictions [6]. In EpiFilter, predictions depend on the product $R_t\Lambda_t$. When the generation time, and consequently $\Lambda_t$, is misspecified, the inferred $R_t$ adjusts in the opposite direction. Hence, its product (predicted incidence) is almost unchanged even if the assumed generation time differs. As multi-step

forecasts are iterated from the 1-step ahead forecast, we expect that the performance of longer forecast horizons will also have negligible differences.

We also assessed how reporting delays and resulting underreporting influenced the real-time estimation of td-$R_t$, ti-$R_t$ and $\Omega_t$. Incorporating the reported delay distribution introduced a similar downward bias across all real-time estimates of td-$R_t$, ti-$R_t$ and $\Omega_t$ (Fig G in S1 File). These results show that while underreporting leads to general underestimation in real-time inference, it affects each estimate in the same way. Nevertheless, td-$R_t$ is likely to provide more accurate real-time inference than ti-$R_t$, as supported by our simulation results.

## Discussion

In this study, we developed a framework to estimate a real time temperature-dependent reproduction number (td-$R_t$) by incorporating a temperature-dependent generation interval into the EpiFilter algorithm. We demonstrated that accounting for temperature-driven variation in the generation interval is essential for accurately estimating the real time reproduction number $R_t$ for dengue. In some cases, excluding temperature from the estimation of $R_t$ could lead to worse performance than the angular reproduction number ($\Omega$) that does not take in any generation time information. We evaluated the performance of td-$R_t$ compared to the temperature-independent reproduction number (ti-$R_t$) and the angular reproduction number ($\Omega$) under real data and under simulation scenarios where the generation interval is temperature-dependent. Our findings show that td-$R_t$ consistently outperforms ti-$R_t$ and $\Omega_t$ in terms of classifying whether transmission is increasing ($R_t>1$) or not, particularly when the generation interval is influenced by temperature. We also found that td-$R_t$ was more concordant with the true $R_t$ than ti-$R_t$ or td-$R_t$ when temperature varied more or when the true, underlying $R_t$ fluctuated more rapidly. Overall, our results underscore the importance of accounting for temperature-driven dynamics in the generation interval, especially for vector-borne diseases like dengue where environmental conditions directly influence transmission processes [23,24]. Our framework is well-suited for real-time analysis as it incorporates temperature in the estimation of $R_t$, allowing the method to capture drivers of transmission that are not directly observed from case data alone. Moreover, the Bayesian recursive filtering step in our framework improves the reliability of real-time estimates as they are conditioned on all past data. In contrast, real-time, sliding-window methods such as EpiEstim are unable to leverage the whole dataset and rely on only a fixed window of recent incidence data. These properties highlight the applicability of our framework for real-time analysis of transmission rates.

We found that incorporating temperature-dependence in the estimation of the reproduction number improved its accuracy in classifying if the epidemic is growing or not. This aligns with the results from Codeço et al, which found that considering the effect of temperature on the generation interval provided a more precise and accurate estimate of the reproduction number [11]. The relationship between temperature and the reproduction number is biologically motivated. In vector-borne diseases like dengue, temperature directly influences the extrinsic incubation period, which is the time taken for a virus to develop within the mosquito before the vector becomes infectious [9]. Higher temperatures typically shorten the extrinsic incubation period, accelerating transmission cycles [9,10]. Our results show that taking into account this aspect of transmission dynamics increased the accuracy and precision of the reproduction number estimate. Expectedly, we found that the relative accuracy of td-$R_t$ was higher when temperature had larger fluctuations. We showed that td-$R_t$ would be particularly useful in regions which are more seasonal, such as upper Northern Thailand where daily temperatures varied from 18°C in the cool season to 29°C in the hot season. This is likely because the larger changes in temperature would result in larger changes in the generation interval. For example, the range of serial intervals in Northern Thailand is 60% larger than in Singapore, a difference that td-$R_t$ is able to specifically account for, unlike ti-$R_t$ or $\Omega$. However, our results showed that even in areas where temperatures are relatively stable like Singapore, td-$R_t$ still maintains an advantage of accuracy.

In our findings from real data, the signals of td-$R_t$ and $\Omega_t$ were highly similar while ti-$R_t$ and $\Omega_t$ were more divergent. This refers to the concordance in identifying whether transmissibility was above or below 1, rather than exact agreement

in the magnitude of estimates. In simulations, td-$R_t$ mostly outperformed $\Omega_t$. This may be because our simulations primarily varied temperature and the underlying $R_t$, allowing td-$R_t$ to leverage temperature information effectively and account for the variation in transmission. While the true Rt in real-world settings is unknown, it is expected to fluctuate due to short-term factors such as human movement and vector control interventions like fogging. We found that in simulation scenarios where random external noise was introduced to the true underlying $R_t$, $\Omega_t$ occasionally had a higher concordance with the true $R_t$ than both td-$R_t$ and ti-$R_t$. However, td-$R_t$ remained the most accurate metric overall. In practice, combining td-Rt, ti-$R_t$, and $\Omega_t$ could leverage the strength of each metric and improve accuracy. One potential method would be to categorise periods of $R_t > 1$ and $R_t \leq 1$ based on a majority vote across the three metrics. This approach could help produce more reliable estimates of transmissibility.

td-$R_t$ performance was reduced when transmission rates were underestimated or overestimated. Overestimation in transmission rates could arise from changes in contact rates due to vector control which influence other components of the generation interval, whereas underestimation could arise from a reduction in vector control that is not incorporated into estimation. For example, the deployment of Wolbachia-infected mosquitoes in Singapore suppresses the mosquito population, reducing the rate of mosquito-human interaction, thereby lengthening the generation interval and potentially lowering $R_t$ [25–29]. In such cases where vector control interventions lead to substantial fluctuations in mosquito population and transmission rate, it may be more appropriate to use td-$R_t$ in combination with ti-$R_t$ or $\Omega_t$ as a measure of transmissibility.

When comparing the $\Omega_t$ and ti-$R_t$, we noted that $\Omega_t$ generally achieved higher AUC-ROC values than ti-$R_t$, whereas ti-$R_t$ achieved a higher percentage accuracy than $\Omega_t$. AUC-ROC evaluates a model's ability to discriminate between epidemic and controlled periods across all possible thresholds of $P(R_t > 1)$. In contrast, percentage accuracy reflects a decision threshold at $P(R_t > 1) = 0.5$, which directly corresponds to the epidemiological threshold of $R_t = 1$. This suggests that while $\Omega_t$ may separate the two conditions more effectively, ti-$R_t$ aligns more closely with the practical decision boundary. While we chose a decision threshold of 0.5, policymakers may set lower or high cutoffs in practice.

This study compared three transmissibility estimates requiring different levels of real-time information. Apart from daily incidence, td-$R_t$ requires the most amount of information (daily temperature, generation interval distribution), followed by ti-$R_t$ (generation interval distribution) and $\Omega_t$ (estimated mean generation time). td-$R_t$ which utilises more information can potentially provide more accurate estimates, while $\Omega_t$ may be more feasible when data is sparse or unavailable.

Our study has several strengths. First, we used a simulation-based framework with known ground truth to systematically evaluate the accuracy and robustness of the reproduction number estimates under a range of realistic conditions, such as varying ranges of temperature and underlying $R_t$. Second, we incorporated both real and synthetic temperature datasets to reflect different levels of environmental variability. Thirdly, we compared the performance of the temperature-dependent reproduction number with competing metrics, including the angular reproduction number, which allowed us to assess its relative accuracy and determine whether td-$R_t$ is a useful alternative to existing frameworks. However, our work also has limitations. Firstly, we used contemporaneous temperature to retrospectively estimate the serial interval distribution. However, if temperatures fluctuated significantly in the days preceding the contemporaneous day, this could introduce measurement error into the estimates. Furthermore, the specification of the serial interval distribution was based on limited biological data, which could be biased. Nevertheless, the high accuracy of the $R_t$ estimates suggests that the chosen specification was sufficiently appropriate for the purposes of this analysis. Furthermore, our approach did not explicitly account of reporting delays as we had access to the date of onset in the Singapore dengue case data (and not just the reporting date). As we also consider the serial interval distribution, which is the time between dates of onset, the incidence of onset times was the most appropriate data source. We found that in the presence of reporting delays, real-time transmissibility estimates may be underestimated. Our methodology can be extended to account for reporting delays by incorporating a nowcasting step to adjust for right truncation bias. This could be done by estimating the expected number of cases that have occurred but have not yet been reported using historical reporting delay distributions and recent incidence patterns, as implemented in frameworks like EpiNow2. Additionally, while we focused on

dengue, the findings may not fully extend to diseases with different transmission mechanisms. For instance, vector-borne zoonotic diseases such as *Plasmodium knowlesi* malaria involve complex transmission cycles that include animal reservoirs (monkeys), vectors (mosquitoes), and humans [30,31]. In these cases, transmission can occur via multiple pathways which complicates the modelling approach. However, this framework could be applied to other diseases by modifying the convolution of generation interval distribution appropriately. Lastly, we were unable to assess the performance of the transmissibility estimates on real data as the ground truth is unknown and estimates of model accuracy relied on simulated data instead.

In conclusion, we showed that explicitly modelling the temperature dependence of the generation interval improves $R_t$ estimation when temperature is a key driver of transmission dynamics, particularly in the presence of strong environmental variability. In noisier contexts where many other unobserved factors may be present, more flexible approaches like the angular reproduction number could be advantageous as a complementary statistic to the temperature-dependent reproduction number. Future work should explore hybrid methods that can dynamically adapt to both structured environmental signals and unstructured noise in transmission data, and validate findings across diverse vector-borne disease settings.

## Supporting information

**S1 File. Appendix containing additional details on results.**
(DOCX)

## Author contributions

**Conceptualization:** Esther Li Wen Choo.

**Data curation:** Esther Li Wen Choo, Jo Yi Chow.

**Formal analysis:** Esther Li Wen Choo.

**Investigation:** Esther Li Wen Choo.

**Methodology:** Esther Li Wen Choo, Kris Varun Parag.

**Project administration:** Jue Tao Lim.

**Resources:** Jue Tao Lim.

**Software:** Esther Li Wen Choo.

**Supervision:** Jue Tao Lim.

**Visualization:** Esther Li Wen Choo.

**Writing – original draft:** Esther Li Wen Choo.

**Writing – review & editing:** Esther Li Wen Choo, Kris Varun Parag, Jo Yi Chow, Jue Tao Lim.

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
