## [Decision Letter · Decision Letter 0]

24 Aug 2025

Measuring real-time disease transmissibility with temperature-dependent generation intervals

PLOS Computational Biology

Dear Dr. Choo,

Thank you for submitting your manuscript to PLOS Computational Biology. After careful consideration, we feel that it has merit but does not fully meet PLOS Computational Biology's publication criteria as it currently stands. Therefore, we invite you to submit a revised version of the manuscript that addresses the points raised during the review process.

Please submit your revised manuscript within 60 days Oct 24 2025 11:59PM. If you will need more time than this to complete your revisions, please reply to this message or contact the journal office at ploscompbiol@plos.org. Please include the following items when submitting your revised manuscript:

We look forward to receiving your revised manuscript.

Kind regards,

Sasikiran Kandula

Academic Editor

PLOS Computational Biology

Roger Kouyos

Section Editor

PLOS Computational Biology

**Additional Editor Comments:**

I share reviewer 1’s concern about the authors’ insufficient engagement with reporting delays/right truncation, and its implications on real-time applicability of the proposed method. The authors should consider updating their approach. Additionally, it would also be good for the authors to indicate whether the simulated data accounts for environmental factors other than temperature that impact transmission (for example, precipitation). This is perhaps related to some of the points raised by reviewer 2 about the representativeness of simulated data. Both reviewers suggest extending the analysis with temperature data from Northern Thailand.

**Journal Requirements:**

At this stage, the following Authors/Authors require contributions: Esther Li Wen Choo, Kris Varun Parag, Jo Yi Chow, and Jue Tao Lim. Please ensure that the full contributions of each author are acknowledged in the "Add/Edit/Remove Authors" section of our submission form.

6) Kindly revise your competing statement to align with the journal's style guidelines: 'The authors declare that there are no competing interests.'

**Reviewers' comments:**

Reviewer's Responses to Questions

**Comments to the Authors:**

Reviewer #1: Summary

This manuscript presents a novel framework for estimating the effective reproduction number (Rt) in real time, incorporating temperature as a covariate. The approach is particularly relevant for vector-borne diseases such as dengue, where transmission is temperature dependent. Compared to prior efforts (e.g., Codeço et al.), the authors' key innovation lies in the real-time implementation of the framework and its evaluation using simulated datasets under a range of Rt (varying in magnitude, smoothness, and periodicity) and temperature patterns. The results indicate that temperature dependent Rt estimates are more robust than those that do not account for temperature effects. The manuscript is clearly written, and the methodology is described in sufficient detail. However, I have several major and minor concerns that should be addressed to strengthen the manuscript.

Major comments

1. Real-time estimation and right truncation bias

The manuscript claims to estimate Rt in real time. However, it does not appear to address right truncation bias due to delays in case reporting, which is a common issue in real-time epidemiological data. Without adjusting for this bias, the most recent Rt estimates are systematically underestimated, which can falsely suggest declining transmission. This undermines the use of the proposed real-time estimation framework in practical applications.

Given that statistical tools for nowcasting and correcting for right truncation are readily available (e.g., EpiNow2), the omission of such adjustments is a critical limitation. I strongly recommend that the authors either incorporate a right truncation correction or explicitly discuss this limitation and its implications for real-time Rt estimation.

2. Scope of temperature variability in simulation scenarios

The differences between the temperature variation scenarios presented appear relatively small. To better demonstrate the benefits of including temperature in Rt estimation, I suggest incorporating an additional scenario with more pronounced seasonal variation, such as temperature data from northern Thailand.

3. Clarification on simulation design

The manuscript refers to 54 sets of simulated epidemic curves (line 222), but these are not introduced earlier. Please provide a clear description of how these simulation scenarios were generated and their purpose within the study design in the method section.

Minor Comments

• Line 61: Consider clarifying that the real-time limitations of the Wallinga–Teunis method can be mitigated by incorporating nowcasting methods to account for reporting delays.

• Line 140: Please clarify whether a 35-day truncation period is sufficient to capture the generation interval across all temperature ranges considered in the analysis.

• Figure 2: The legend refers to a "blue label," which does not appear clearly in the figure. Please revise or clarify.

• Lines 218–219: Provide additional detail on the observed increase in variance. Articulate more details would help the reader better understand the impact.

Conclusion

This is a well-structured and timely manuscript addressing an important methodological gap in real-time Rt estimation for vector-borne diseases. However, the omission of adjustments for right truncation in real-time case data limits the applicability of the framework for real-world use. Addressing this issue, either methodologically or through a transparent discussion, would substantially strengthen the study. I recommend major revisions before the manuscript can be considered for publication.

Reviewer #2: review is uploaded as a Word document

**Have the authors made all data and (if applicable) computational code underlying the findings in their manuscript fully available?**

Reviewer #1: **No:** Codes used for the analyses is not available

Reviewer #2: Yes

PLOS authors have the option to publish the peer review history of their article (what does this mean? ). If published, this will include your full peer review and any attached files.

**Do you want your identity to be public for this peer review?** For information about this choice, including consent withdrawal, please see our Privacy Policy .

Reviewer #1: No

Reviewer #2: No

**Figure resubmission:**
---

## [Decision Letter · Decision Letter 1]

7 Nov 2025

PCOMPBIOL-D-25-01284R1

Measuring real-time disease transmissibility with temperature-dependent generation intervals

PLOS Computational Biology

Dear Dr. Choo,

Thank you for submitting your manuscript to PLOS Computational Biology. After careful consideration, we feel that it has merit but does not fully meet PLOS Computational Biology's publication criteria as it currently stands. Therefore, we invite you to submit a revised version of the manuscript that addresses the points raised during the review process.

We look forward to receiving your revised manuscript.

Kind regards,

Sasikiran Kandula

Academic Editor

PLOS Computational Biology

Roger Kouyos

Section Editor

PLOS Computational Biology

**Additional Editor Comments:**

The manuscript has been substantially updated and I believe most of the reviewers' concerns have been addressed. Reviewer 2 has follow-up questions and suggestions which I hope the authors can consider.

I reviewed the authors' responses to reviewer 1, who was unavailable. I believe the authors changes are adequate, with a possible exception in their response to comment #1 -- I am unsure how data being 'arranged' by onset date assures effect of reporting delays to be 'likely minimal', since data could be reported at a later date with the correct onset date, and thus an issue for real-time estimation. Perhaps further clarification on what the authors mean (or whether this assertion has been tested) and/or updates to relevant text in the manuscript could be useful.

Additionally, the authors should consider more careful copy editing throughout to improve comprehension.

**Reviewers' comments:**

Reviewer's Responses to Questions

**Comments to the Authors:**

Reviewer #2: Thank you for your thorough responses to my comments. I have just a few follow-up questions:

Lines 87-92: Thank you for the clarification and additional details here. However, I still find this section confusing – you mention that EpiFilter can be used in real time, but then discuss a smoothing step that uses future incidence, which shouldn’t be available in real-time. In a real-time forecast, would only the filtering step be used?

Lines 290-296: Could you specify by how much the generation interval is over- or underestimated in each of the four groups?

Lines 264-271: Thank you for clarifying this. In addition to magnitude, are the simulations also realistic in terms of any observed seasonality in dengue cases? How many outbreaks are expected within a 200 day period of time?

Lines 419-427: I’m still a bit surprised about the similarity in error between these predictions. You say that predictions are insensitive to the generation time, but here, aren’t you estimating a reproductive number, then forming a corresponding forecast? Wouldn’t we expect the forecast generated from an Rt > 1 to be quite different than one generated with Rt < 1, as one would tend to produce an increase in cases and one a decrease? For the same reason, I would also tend to expect the difference in forecasts to compound over longer forecast horizons.

Line 495: Instead of td-Rt, should this say ti_Rt?

Lines 444-450: While it makes sense that, by taking temperature into account, you are not relying only on case counts, I still wonder to what extent underreporting and underestimation of cases in current and recent weeks might affect real-time estimation of Rt. Is this something you could test – for example, by misspecifying the number of cases with symptom onset in recent weeks to represent underreporting, then testing whether you get the same results?

**Have the authors made all data and (if applicable) computational code underlying the findings in their manuscript fully available?**

Reviewer #2: Yes

PLOS authors have the option to publish the peer review history of their article (what does this mean? ). If published, this will include your full peer review and any attached files.

**Do you want your identity to be public for this peer review?** For information about this choice, including consent withdrawal, please see our Privacy Policy .

Reviewer #2: No

**Figure resubmission:**
---

## [Decision Letter · Decision Letter 2]

5 Dec 2025

Dear Ms Choo,

We are pleased to inform you that your manuscript 'Measuring real-time disease transmissibility with temperature-dependent generation intervals' has been provisionally accepted for publication in PLOS Computational Biology.

Best regards,

Sasikiran Kandula

Academic Editor

PLOS Computational Biology

Roger Kouyos

Section Editor

PLOS Computational Biology

Reviewer's Responses to Questions

**Comments to the Authors:**

Reviewer #1: Thank the authors to address my concerns. Regarding to the real-time analysis comments, I think the authors have sufficiently address them in the discussion without the need of extra analyses.

The authors also addressed my other concerns well.

Reviewer #2: Thank you for your thorough responses to my comments. I have no further feedback.

**Have the authors made all data and (if applicable) computational code underlying the findings in their manuscript fully available?**

Reviewer #1: Yes

Reviewer #2: Yes

PLOS authors have the option to publish the peer review history of their article (what does this mean? ). If published, this will include your full peer review and any attached files.

**Do you want your identity to be public for this peer review?** For information about this choice, including consent withdrawal, please see our Privacy Policy .

Reviewer #1: No

Reviewer #2: No

---

## [Editor Report · Acceptance letter]

PCOMPBIOL-D-25-01284R2

Measuring real-time disease transmissibility with temperature-dependent generation intervals

Dear Dr Choo,

I am pleased to inform you that your manuscript has been formally accepted for publication in PLOS Computational Biology. Your manuscript is now with our production department and you will be notified of the publication date in due course.

With kind regards,

Anita Estes
